# Predictability and Effectiveness of Nuvola^®^ Aligners in Dentoalveolar Transverse Changes: A Retrospective Study

**DOI:** 10.3390/biomedicines11051366

**Published:** 2023-05-05

**Authors:** Angelo Michele Inchingolo, Sabino Ceci, Giovanni Coloccia, Daniela Azzollini, Giuseppina Malcangi, Antonio Mancini, Francesco Inchingolo, Paolo Trerotoli, Gianna Dipalma, Assunta Patano

**Affiliations:** Department of Interdisciplinary Medicine, University of Bari “Aldo Moro”, 70124 Bari, Italy; s.ceci@studenti.uniba.it (S.C.); giovanni.coloccia@gmail.com (G.C.); daniela.azzollini93@gmail.com (D.A.); giuseppinamalcangi@libero.it (G.M.); dr.antoniomancini@gmail.com (A.M.); paolo.trerotoli@uniba.it (P.T.); giannadipalma@tiscali.it (G.D.); assuntapatano@gmail.com (A.P.)

**Keywords:** dentistry, orthodontics, removable orthodontics appliances, tooth movement techniques, corrective orthodontics, dental technology, malocclusion, orthodontic appliance design

## Abstract

Nowadays, many people use clear aligners to address their dental issues. The efficacy of transparent dental aligners must be investigated even though they are more aesthetically pleasing, easy to use, and tidy than permanent tools. Thirty-five patients in this study’s sample group who used Nuvola^®^ clear aligners for their orthodontic therapy were prospectively observed. Initial, simulated, and final digital scans were analysed with a digital calliper. The actual results were compared with the prescribed ending position to evaluate the efficacy of transversal dentoalveolar expansion. Aligner treatments in Groups A (12) and B (24), particularly in the dental tip measures, demonstrated high adherence to the prescription. On the other hand, the gingival measures exhibited a greater level of bias, and the differences were statistically significant. However, there was no difference in the outcomes between the two groups (12 vs. 24). Within specific parameters, the evaluated aligners were shown to be helpful in predicting movements in the transverse plane, particularly when considering movements linked to the vestibular–palatal inclination of the dental elements. This article compares the expansion effectiveness of Nuvola^®^ aligners compared with other work in the literature using competitor companies.

## 1. Introduction

In 1946, Harold Dean Kesling proposed the idea of gradually realigning crooked teeth using a series of thermoplastic tooth positioners, opening the door to using clear thermoformed orthodontic appliances [1]. While the Food and Drug Administration’s 1998 permission from Align Technology (Align Technology, Santa Clara, CA, USA) to use Invisalign for orthodontic treatment has been seen as the official beginning of the clear-aligner era, this technology has undoubtedly been widely used, and some other brands have been born and developed over the years [2,3]. The desire for orthodontic appliances that are more aesthetically pleasing and more comfortable than traditional fixed appliances has recently increased as more adult patients require orthodontic treatment [4,5]. From a periodontal perspective, using removable devices instead of fixed-appliance treatments improved the periodontal health and plaque index [6].

Nuvola^®^ aligners [7] are made of polyethylene terephthalate glycol [8], whereas other competitive devices, such as Invisalign, are made of polyurethane [9]. The two materials have different mechanical properties, e.g., concerning the elastic modulus, and due to the different chemical compositions. Nevertheless, their effects on the clinic still need to be investigated more. The mean thickness of the main available aligners in the market is 0.75 mm with a different gap (i.e., “fit”) between the aligner and the teeth surfaces, as Lombardo et al. evaluated [10].

Comparing traditional fixed treatments with aligner therapy, it can be seen that they are both valuable in resolving malocclusion, with the advantages of aligners being able to perform isolated or segmented movements and reducing the duration of treatment. However, clear aligners are not effective in achieving optimal occlusal contacts or root torque changes [11].

Only a few studies have concentrated on the predictability of orthodontic tooth movement, even though aligners have been regarded as a safe, aesthetically pleasing, and pleasant orthodontic technique for adult patients [12,13,14]. While initial investigations into the effectiveness of Invisalign clear aligners showed a mean accuracy—defined as the overlap between the projected and accomplished movements—of 41% overall, more recent studies demonstrated that the “gap” between expected movement and actual movement had been reduced in non-extraction orthodontic cases [15]. The predictability of clear-aligner therapy is still very much disputed, particularly regarding complicated tooth movements [16,17]. While it has been shown that levelling and aligning, intrusion, and physical distalization of upper molars (less than 1.5 mm) are predictable movements, rotations, extrusions, and torque movements are challenging to accomplish with clear aligners [18,19,20]. These results differ in studies involving other clear-aligner brands than Invisalign [21,22,23]. 

Houle et al. [24] evaluated the predictability of transverse changes with the Invisalign aligner system by investigating the results in a group of 64 adults. The researchers adopted a method similar to the one presented in this paper, measuring transverse diameters at tooth and gingival points at the canine, premolar, and molar levels. The results they obtained were predominantly coronal tipping movements with a fairly high prescription adherence to the final result. However, the results became less accurate moving from the anterior to the posterior portion. The simulation software, however, overestimated the expansion and simulated a more body-like movement than was actually achieved.

In contrast, Solano-Mendoza et al. had conflicting results in their work with 116 patients where the predictability of expansive movement with Invisalign Ex30 material was assessed. However, this material is different from that used in the other studies, which were conducted on the Smart Track. In fact, it has a thickness of 0.030 inch in a single layer. The authors evaluated not only the width at the canine, premolar, and molar levels, but also the rotations and inclinations. The results were, as far as expansion is concerned, that Clincheck was not predictable because there were statistically significant differences.

This work aims to compare the cross-sectional measurements of Nuvola^®^ simulation software with the final clinical result and evaluate the degree of dentoalveolar expansion achieved with Nuvola^®^ aligners on the maxillary arch.

## 2. Materials and Methods 

The sample of 35 digital upper dental arch impressions of patients that underwent orthodontic treatment with clear aligners was provided by Nuvola^®^ (Nuvola World Srl, Via L.L. Zamenhof, 615 36100—Vicenza, Italy) and divided into groups: Group A, composed of 10 patients, and Group B, composed of 25 patients. Each model was provided at *t*_0_ (pre-treatment), *t*_1_ (post-treatment), and final digital setup (s1). Group A was treated with 12 aligners, while Group B was treated with 24. The movement was programmed with simultaneous modification on all elements that changed every 14 days. Each patient was assigned a number to identify with respect to personal data. To measure the outcome, digital prediction casts were also obtained from the Nuvola^®^ simulation software and were compared with casts after 12 or 24 aligners. No other data about the patients were available. The analysis of orthodontic models was conducted by two trained orthodontics specialisation school students at the University of Bari “Aldo Moro” using the Medit Link software base and Medit Design app (Medit corp. 23 Goryeodae-ro 22 gil, Seongbuk-gu, Seoul, Republic of Korea). Using the digital calliper present within the Medit Design app, the students recorded the linear values of the widths of the jaw arches considering the tips of the cusps and the most palatal points of the gingival margin of the canines, premolars, and first molars, as shown in Figure 1 [24]. In particular, the cusps considered for tooth position were the palatal cusps of the premolars and the mesiopalatal cusps of the first molars. At the same time, the gingival points individuated were the most palatal margins of the teeth (Figure 2). The points identified for each tooth considered (cusp apex and gingival margin) were chosen to better represent transverse tipping and bodily movement, respectively. Only patients with fully erupted elements were considered in this study. 

### Data Analysis

Three comparisons were made for each type of measurement: 

The programmed movement is the difference between the simulation *s*_1_ and pre-treatment values *t*_0_ to evaluate the amount of programmed movement.
$$Programmed\;Movement=s_{1}-t_{0}$$

The observed movement is the difference between the values at the end of the treatment *t*_1_ and the initial ones *t*_0_ to evaluate the amount of movement obtained.
$$Observed\;Movement=t_{1}-t_{0}$$

The imprecision, which is the difference between the ideal values of the simulation and the real post-treatment ones, was used to identify the difference between the post-treatment positions obtained and those programmed by the software.
$$Imprecision=s_{1}-t_{1}$$

The accuracy was defined as the ratio between the imprecision and the observed movement, and was expressed as a percentage:$$Accuracy=\frac{Imprecision}{Observed\;Movement}\times 100$$

The Shapiro–Wilks test was applied to evaluate the normal distribution of data. Because all variables approach the normal distribution, the measures were summarised as mean and standard deviation, and the parametric statistic was applied to evaluate differences and relationships.

A repeated-measures Analysis of Variance (ANOVA) model was applied to evaluate the difference in prescription and observed movement between Group A and Group B. For each *i*-th subject, the model was:Movement_ijklm_ = μ + α_j_ + β_k_ + θ_l_ + δ_m_ + 
                                           + α_j_β_k_ + α_j_θ_l_ + α_j_δ_m_ + θ_l_δ_m_+
                   + β_k_θ_l_ + β_k_δ_m_ +
                                                                                   + α_j_β_k_θ_l_+ α_j_β_k_δ_m_+ α_j_θ_l_δ_m_ + α_j_ β_k_ θ_l_ δ_m_ + ε_ijklm_
where variables included in the model were as follows:μ is the intercept; ε is the residual;α is the aligners’ duration of treatment, with j = 1,2, which are Group A and Group B;β is the point, with k = 1,2, which are tip and gingival;θ is the tooth, with l = 1,2,3,4, which are canine, first premolar, second premolar, and first molar;δ is the type of measure, with m = 1,2, which are the simulated or post-treatment measures. 

Aligners were treated as independent factors, while all others were considered intra-individual correlated measures, and the type of measure was considered as a repeated measure. The *p*-values of the post hoc comparisons were adjusted using a step-down Bonferroni method. 

The Bland and Altman plot was applied to evaluate the agreement between predicted and post-treatment measures. The bias was the difference between predicted change and post-treatment change. The x-axis of the plots was the actual post-treatment change; the y-axis was the bias. The limits of agreement were determined as the mean plus/minus (upper/lower limit) 1.96-fold the standard deviation. A *p*-value < 0.05 was considered for statistical significance. All analyses were performed using Statistical Analysis System (SAS) 9.4 for personal computer.

## 3. Results

The analysis was performed with a single ANOVA model for repeated measures with a hierarchical structure; therefore, in a single model, all factors were evaluated that could contribute to the changing of the distances. The model resulted in a statistically insignificant effect for the group (F = 2.81, *p* = 0.1029), which was not significant either in interaction terms with the tooth (F = 0.51, *p* = 0.59) or in the type of measurement (F = 0.6, *p* = 0.4449). There was a statistically significant difference due to the dental or gingival measures (F = 4.59, *p* = 0.0396) and for the tooth (F = 13.24, *p* < 0.0001). The difference of movement between Groups A and B did not show statistically significant results among the subgroups of point, teeth, and type of measures (F = 0.9, *p* = 0.4456).

The comparison between predicted and observed changes in the distance is described in Table 1 for Group A and Table 2 for Group B. After the adjustment of *p*-values in Group A, there were statistically significant differences for gingival measures of the first premolar (2.5 vs. 1.5, *p* = 0.048), second premolar (2.8 mm vs. 1.6 mm, *p* = 0.048), and first molar (2.4 mm vs. 1.4 mm, *p* = 0.0286).

The Bland and Altman graph to evaluate agreement among measures in Group A (Figure 3 and Figure 4) allows for some considerations. Dental measures have shown a mean difference not overlapping the zero bias, especially for the second premolar, whose mean bias (standard deviation—sd) was 0.55 (0.81); the molar measures, instead, have shown a significant standard deviation of 2.78. In the first case, it is possible to observe a significant bias, even if each measure is within the limit of agreement; in the second case, the limits of agreement have a large width, and it appears that data are included in the limits of agreement, but this happens in a wide range. Furthermore, if the relation between the true measure and the bias predicted was analysed in both cases, a linear trend could be seen. For the second premolar, the trend was not statistically significant. In contrast, for the molar tip, the trend was statistically significant with a regression coefficient = −1.22 (*p*-value = 0.0043), suggesting that the agreement decreases by 1.22 mm for each 1 mm difference between the observed and simulated increases. Other dental measures in Group A did not show particular bias or trends in bias.

The agreement in gingival measures (Figure 4) was larger for each point: the lower mean bias (sd) was observed in the canine with a bias of 0.58 (1.13), while the larger in the second premolar resulted in 1.13 (0.94). The limits of agreement included the zero bias, but it was not very near to the mean bias; in any case, there was not a trend in bias in any tooth. 

The differences between observed and simulated measures in Group B were statistically significant only for the gingival measures: for the first premolar it was 3 mm vs. 2.1 mm (*p* = 0.0252), for the second premolar 3.6 mm vs. 2.2 mm (*p* = 0.0016), and for the first molar 3.4 mm vs. 1.6 mm (*p* = 0.0045). In this group, the Bland and Altman plots showed significant wide-ranging biases in the limits of agreement. In the canine dental measure (Figure 5), the limits of agreement were wide because there was a significant standard deviation even with a small mean bias (sd) of 0.18 mm (1.39). Secondly, two canine dental measures were out of the limits of agreement, and a statistically significant trend was observed: a decrease of 0.32 mm in the difference between observed and simulated measures for each 1 mm increase in the observed change in the measure (*p* = 0.02). Mean biases (sd) for other teeth were 0.76 mm (1.34) for the first premolar, 0.57 mm (1.61) for the second premolar, and 0.83 mm (1.52) for the molar. The more considerable bias was for the first premolar, but the standard deviations determine wide limits of agreement, and in each tooth, there was at least one measure outside the limits.

The results for gingival measures (Figure 6) showed more significant mean biases and broader limits. The canine bias (sd) was 1.18 (2.28), and there were two teeth with results outside the limits of agreement; the first premolar, the tooth with the lower bias, had a mean bias (sd) of 0.84 (1.19), and no teeth outside the limits; the second premolar showed a mean bias (sd) of 1.42 mm (1.58) and one tooth outside the limits of agreement; and the molar showed a mean bias of 1.79 (2.15), the largest values of all biases measured, and two teeth outside the limits of agreement. In any case, the trend was statistically significant.

With reasonable doubt, the aligner treatment in Group A (12) and Group B (24) shows a good degree of adherence to the prescription, especially in the dental tip measures. On the other hand, the gingival measures show a higher degree of bias, and the differences were statistically significant. The effects, however, did not differ between the two groups (12 vs. 24). A summary diagram of the study design and average results is shown in Figure 7.

## 4. Discussion

This study is about the effectiveness of clear aligners in expansion movements. Understanding how much of the planned expansion is achieved by measuring the lateral–posterior teeth distance is helpful. In addition, the landmarks that should be simple to detect have to indicate the direction of expansion, so more crown tipping translates to a vestibular direction involving the roots. Although the models of the final setup do not always show a perfect alignment of the arches, they are indicators of what has been designed. They can be taken as a reference to understand how much they differ from the position that the teeth have reached in vivo. Easily identifiable landmarks, such as cusps and palatal gingival points, were used to measure changes in width from the beginning to the end of treatment, considering the virtual setup. The points identified for each tooth best represent the transverse tipping and body movement, respectively.

Clear orthodontic aligners were initially developed mainly to address small tooth position anomalies. The correction of minor malocclusions in adolescents demonstrated that transparent aligners were as effective as fixed appliances, with significantly better outcomes for tooth alignment [25]. Still, orthodontists frequently need more time to plan clear-aligner treatment than fixed-appliance therapy [26,27]. While some claim to focus on complex malocclusions, specific aligner systems remain purposefully and openly confined to repairing minor positional anomalies [28]. According to a systematic review by Rossini et al. [6], invisible aligner therapy allows predictable intrusion and distalization movements of posterior sectors up to 1.5 mm. Regarding posterior vertical control movements, anterior extrusion, and rotation of rounded teeth, there are no satisfactory results. In addition, they suggest the non-isolated use of invisible aligners, but also of auxiliary devices such as intermaxillary elastics, customised attachments, and interproximal reduction.

### 4.1. The Expansion Problem

Expansion is one of the most common orthodontic treatments that create space to align teeth in the upper and lower arches [29,30,31]. Other therapeutic strategies exist to achieve arch space so that alignments and crowding can be resolved. These solutions often complement each other by composing a more structured treatment plan. These include interproximal reduction, proinclination, and in some cases, distalization [32]. Interproximal reduction cannot be performed indiscriminately but must adhere to the limits of interdental enamel removal. According to Zachrisson, there is no scientific evidence for an increased risk of damage to dental or periodontal structures in cases of interproximal reduction [33]. As Bishara [28] stated, the intercanine and intermolar widths tend to increase from the age of 3 to the age of 13 both in the lower and upper jaw and do not increase more during later years. Ageing tends to contract the arches over the years as the dental arches tend to increase dental crowding [34,35]. On the contrary, there is a slight decrease in arch perimeter over the years with a reduction in transversal width. Most treatments in adults with clear aligners aim to align teeth using the expansion of the arches and interproximal reduction. Clear aligners during expansion produce force direction applied far from the centre of resistance of the teeth, so even if the aligner treatment is planned to involve a transverse expansion, the orthodontic pressure tends to generate coronal tipping and some “flaring” of the teeth. This tipping often must be controlled [29,36]. If root–vestibular torque is added during the development of the virtual setup in the lateral–posterior segments and thick, retentive vestibular attachments are placed, movement is facilitated, and aligners have a better fit [37,38]. When studying and testing the treatment simulation, it is essential to consider the shape of the attachments, as these can influence the retention of the aligners. It is believed that attachments have a significant impact on improving retention [39].

Few studies have evaluated simulated dental expansion with the final result of orthodontic treatment [6,21,22]. Moreover, other parameters that must be considered are the thickness and quality of the materials and the amount of movement generated with each aligner [10]. Lione et al. [40] evaluated teeth movements during maxillary expansion with aligners. The Invisalign system [41] produces arch extension with minimal buccal tipping. However, there were considerable variations between planned and actual maxillary canine outcomes. The growth rate of the maxillary arch decreased gradually in the canine, premolar, and posterior regions, with the most significant net increase at the first and second premolars [40,42,43]. Boyd and Waskalic [44] evaluated the treatment of different complex malocclusion problems using Invisalign clear aligners. They stated that buccal expansion of 2 to 4 mm could be achieved using this device, helping crowded anterior crowding resolution or arch morphology changes. Although most of these changes are probably linked to tipping movement, technicians may be asked to perform the movement as bodily as possible, often also using overcorrection. Perrotti et al. [45] found good efficacy with the Nuvola^®^ Orthodontic Protocol (OP) system in maxillary expansion. However, they adopted a protocol that involved the simultaneous use of a morpho-functional device for 30 min per day. Nevertheless, further studies on this combined use must be carried out to validate its effectiveness. In a paper by Vidal-Bernárdez et al. [46], it was highlighted that invisible aligners, particularly Invisalign, could predictably determine a moderate expansive movement (2–3.5 mm) in both the upper and lower arches. They, too, found that expansion is more predictable at the coronal than at the gingival level. This is in line with the data obtained in the present work. A recent systematic review by Robertson et al. [47] evaluating the effectiveness of invisible aligner therapy showed that there is only one paper that reports that the predictability inherent in transverse dimension changes is more significant in the lower jaw (95.97%) than in the maxilla (77–78%). Furthermore, the review clarifies that many tooth movements are not predictable enough with aligner therapy except for reduced tooth movements in the horizontal plane. In agreement with Lione et al. [40], the data obtained in the present study also show that expansion is linked to an increase in the buccal inclination of the dental elements. However, it is important to emphasise that overcorrection is a tool for making the expected result more likely to be the result planned in the simulation phase.

### 4.2. Compliance and Post-Treatment Retention

It is important to remember that clear aligners are removable appliances, so they are not compliance-free. That variable strongly affects the results. The movements are impossible to gain if the patient does not appropriately apply clear aligners [4,48,49]. The arch width augmentation is difficult to stabilise, so it is necessary to follow some retention program during life to contrast later contraction of the arches. Crucial in these treatments with removable devices is patient compliance. It is possible to incentivize device use and adherence to the prescribed hours (at least 20–22 h per day) by using monitoring tools or digital reminders, as was highlighted by some authors. This digital reminder consists of software for mobile devices that has resulted in improved aligner compliance [48,50]. For example, Invisalign has introduced an app, My Invisalign (Align Technology), that reminds patients when to replace their aligners and also of how many minutes the aligners have been removed from the mouth.

There are various post-orthodontic removable retainer devices on the market, and the most widely used are Essix [51,52] and other invisible thermoplastic retainers [53,54,55]. Based on their clinical experience, the authors suggest using clear retainers overnight to avoid orthodontic recurrences at the end of therapy or the placement of fixed retainers on the lingual and palatal surfaces of anterior teeth.

## 5. Conclusions

The evaluated aligners have proven useful, within certain limits, in determining the predicted movements in the transverse plane, especially considering movements related to the vestibular–palatal inclination of the dental elements. Effectiveness is not related to the number of aligners used, but rather to the location of the expansion. Bodily movements, assessed by gingival point analysis, are significantly less predictable. The simulations represent a valuable tool for the clinician in the visual planning of anticipated movements, although expansive movements should be cautiously estimated. 

The limitations of this study were mainly related to the limited sample size and heterogeneity of patients. The prescriptions or features (attachments, power ridge, etc.) introduced in the individual patients’ treatment plans, apart from the dental expansion values, were unknown. However, it must be borne in mind that these types of orthodontic studies are based on long-term clinical results. Therefore, even if the sample is limited, it is constantly monitored over time and motivated toward therapy, which is fundamental to the predictability of results in this orthodontic treatment method.

Positive aspects of the study are, above all, determined by the fact that there are few works on Nuvola^®^ aligners in the literature, whereas there are more prominent works on other companies’ products. Therefore, this opens a line of research to see if any techniques or indications can be introduced into prescriptions to make expansive movement more predictable and efficient. It would also be nice to show, through cone-beam computed tomography studies, changes in root torque and not just change at the coronal level. However, this raises ethical issues about unnecessary exposure to radiation for non-diagnostic research purposes.

Future perspectives may involve evaluating whether more predictable results can be obtained by adopting different attack characteristics or experimental protocols for expansion. Another aspect that could be evaluated is the use of different materials in the compound that makes up the plastic of the aligners in generating elastic forces of differing intensity at different points.

## Figures and Tables

**Figure 1 biomedicines-11-01366-f001:**
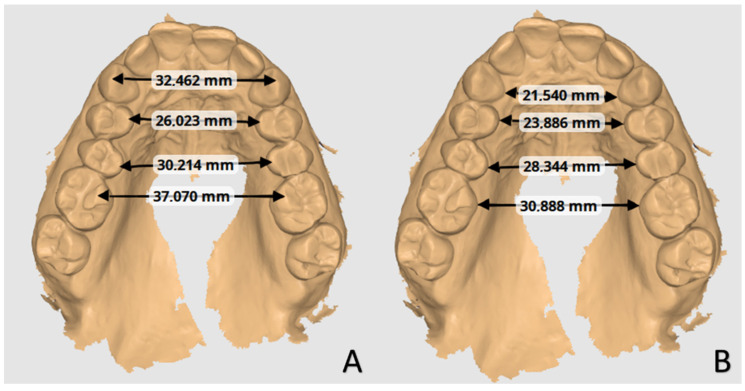
Linear measurements between selected points on sample digital model of the study using the Medit Design app in Medit Link software (Medit corp. 23 Goryeodae-ro 22 gil, Seongbuk-gu, Seoul, Republic of Korea): canine tip, premolar, and molar palatal cusp tip (**A**); canine, premolars’, and first molars’ more palatal gingival margin points (**B**).

**Figure 2 biomedicines-11-01366-f002:**
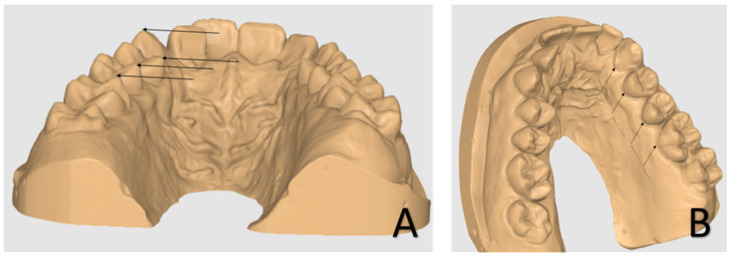
Landmarks—palatal cusp tips (**A**) and most gingival point of the palatal surface of the crown (**B**).

**Figure 3 biomedicines-11-01366-f003:**
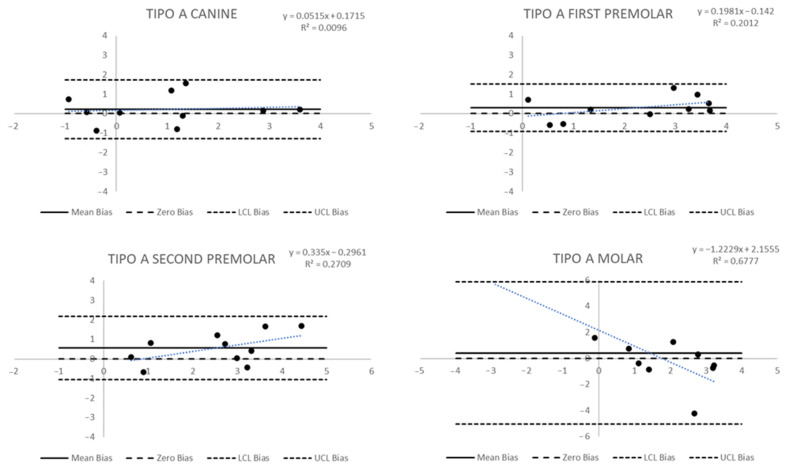
Bland and Altman plots to evaluate the agreement between predicted and post-treatment measures of dental tip changes in Group A.

**Figure 4 biomedicines-11-01366-f004:**
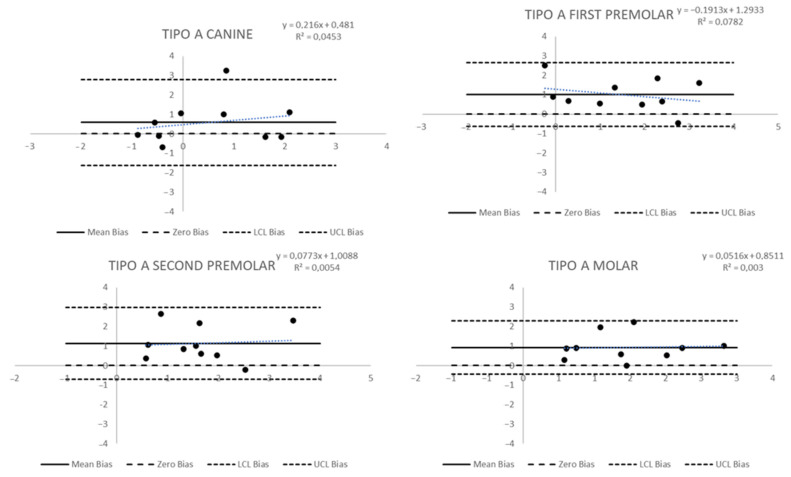
Bland and Altman plots to evaluate the agreement between predicted and post-treatment measures of gingival changes in Group A.

**Figure 5 biomedicines-11-01366-f005:**
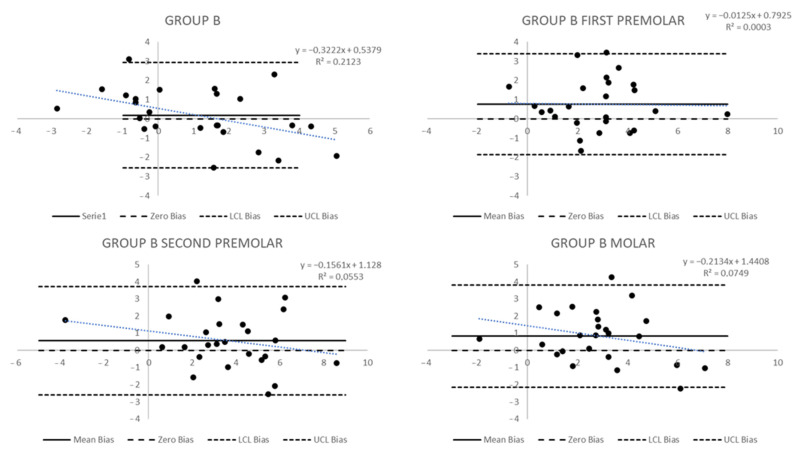
Bland and Altman plots to evaluate the agreement between predicted and post-treatment measures of dental tip changes in Group B.

**Figure 6 biomedicines-11-01366-f006:**
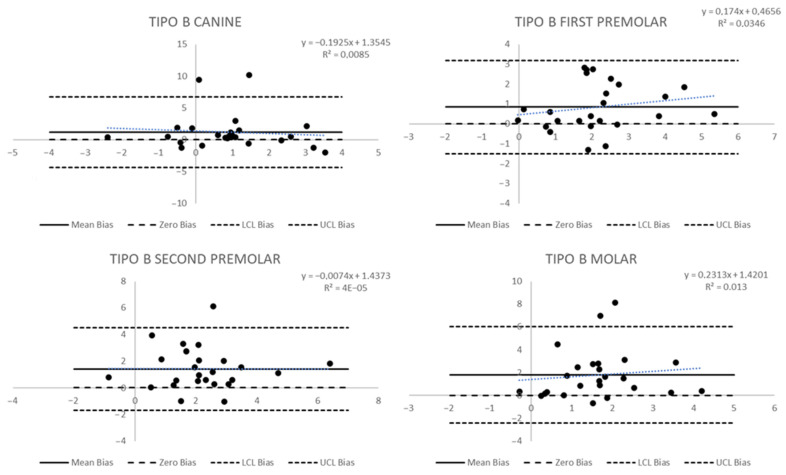
Bland and Altman plots to evaluate the agreement between predicted and post-treatment measures of gingival changes in Group B.

**Figure 7 biomedicines-11-01366-f007:**
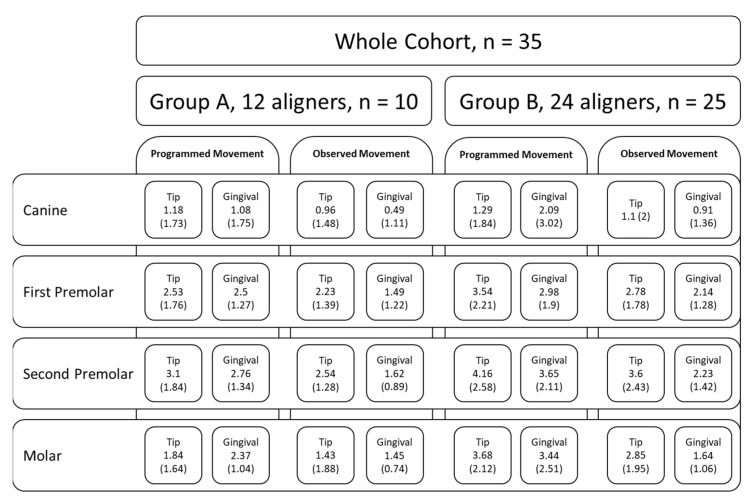
Summary diagram of the study design and average results, with standard deviation in brackets, for each tooth and point considered (tip and gingival). None of the comparisons were statistically significant.

**Table 1 biomedicines-11-01366-t001:** Predictability of changes for maxillary measurements in Group A (12).

Tooth Type	Prescription Change, mm	Post-Treatment Change, mm	Simulation vs. Post-Treatment		Accuracy of Change, %
	Mean (sd)	Mean (sd)	Mean (sd)	Adj *p*-value	
Canine tip	1.2 (1.7)	1.1 (1.5)	0.22 (0.8)	1	20.0
First premolar tip	2.5 (1.8)	2.2 (1.4)	0.3 (0.6)	0.6745	13.6
Second premolar tip	3.1 (1.8)	2.5 (1.3)	0.55 (0.8)	0.4263	22.0
First molar tip	1.8 (1.8)	1.4 (1.9)	0.4 (2.8)	1	28.6
Canine gingival	1.1 (1.7)	0.5 (1.1)	0.58 (1.1)	0.6745	116.0
First premolar gingival	2.5 (1.3)	1.5 (1.1)	1 (0.8)	0.048	66.7
Second premolar gingival	2.8 (1.3)	1.6 (0.9)	1.13 (0.9)	0.048	70.6
First molar gingival	2.4 (1)	1.4 (0.7)	0.93 (0.7)	0.0286	66.4

sd: standard deviation.

**Table 2 biomedicines-11-01366-t002:** Predictability of changes for maxillary measurements in Group B (24).

Tooth Type	Prescription Change, mm	Post-Treatment Change, mm	Simulation vs. Post-Treatment		Accuracy of Change, %
	Mean (sd)	Mean (sd)	Mean (sd)	*p* Value	
Canine tip	1.3 (1.8)	1.1 (2)	0.18 (1.39)	1	16.4
First premolar tip	3.5 (2.2)	2.8 (1.8)	0.75 (1.33)	0.092	26.8
Second premolar tip	4.2 (2.6)	3.6 (2.4)	0.57 (1.61)	0.5412	15.8
First molar tip	1.8 (1.6)	2.8 (2)	0.83 (1.52)	0.1035	29.6
Canine gingival	2.1 (3)	0.9 (1.4)	1.18 (2.82)	0.3832	131.1
First premolar gingival	3 (1.9)	2.1 (1.3)	0.84 (1.19)	0.0252	40.0
Second premolar gingival	3.6 (2.1)	2.2 (1.4)	1.42 (1.57)	0.0016	64.5
First molar gingival	3.4 (2.5)	1.6 (1.1)	1.79 (2.15)	0.0045	111.9

sd: standard deviation.

## Data Availability

Not available.

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
