# Peer review of "Predictability and Effectiveness of Nuvola® Aligners in Dentoalveolar Transverse Changes: A Retrospective Study"

_biomedicines, 2023, doi:10.3390/biomedicines11051366_

Round 1
Reviewer 1 Report
Please see the attachment

Reviewer 2 Report
Reviewer report on Manuscript Draft ‘Predictability and Effectiveness of Nuvola® Aligners in Den-2 toalveolar Transverse Changes: A Retrospective Study’.
This work aimed to compare the cross-sectional measurements of Nuvola® simulation software with the final clinical result and evaluate the degree of dentoalveolar expansion achieved with Nuvola® aligners on the maxillary arch. Therefore, in this research Nuvola® clear aligners for their orthodontic therapy were prospectively observed. Real results were compared to the prescribed ending position to evaluate the efficacy of transversal dentoalveolar expansion. Aligner treatments in groups A (12) and B (24), particularly in the dental tip measures, demonstrate high adherence to the prescription. On the other hand, the gingival measures exhibit a greater level of bias, and the differences were statistically significant. However, there was no difference in the outcomes between the two groups (12 vs 24). Within certain parameters, the evaluated aligners had shown to be helpful in predicting movements in the transverse plane, particularly when considering movements linked to the vestibular-palatal inclination of the dental elements.
The evaluated aligners have proven useful, within certain limits, in determining the 321 predicted movements in the transverse plane especially considering movements related 322 to the vestibular-palatal inclination of the dental elements.
This manuscript is in the scope of journal, it is rather well written and interestingly addressed. Manuscript contributes to the field of analytical chemistry. Therefore, the manuscript can be published after some minor improvements:
Conclusions are not very sound, therefore, this section could be advanced and extended.
‘Future perspectives’, could be presented in Conclusion section instead of being presented in chapter ‘ 4.3. Limits, positive aspects and future perspectives ‘.
N/A
Round 2
Reviewer 1 Report
Paper can be accepted in its present form.
Minor English grammar editing required for the whole manuscript.